# Perfluoroalkyl Carboxylic Acids Interact with the Human Bile Acid Transporter NTCP

Melissa J. Ruggiero [1], Haley Miller [1], Jessica Y. Idowu [1], Jeremiah D. Zitzow [2], Shu-Ching Chang [2] and Bruno Hagenbuch [1,*]

[1]   Department of Pharmacology, Toxicology and Therapeutics, The University of Kansas Medical Center, Kansas City, KS 66160, USA; Melissa.ruggiero@labcorp.com (M.J.R.); haley.miller@drake.edu (H.M.); jidowu@kumc.edu (J.Y.I.)

[2]   Medical Department, 3M Company, St Paul, MN 55144, USA; jzitzow@mmm.com (J.D.Z.); s.chang@mmm.com (S.-C.C.)

*   Correspondence: bhagenbuch@kumc.edu; Tel.: +1-913-588-0028

**Abstract:** Na$^+$/taurocholate cotransporting polypeptide (NTCP) is important for the enterohepatic circulation of bile acids, which has been suggested to contribute to the long serum elimination half-lives of perfluoroalkyl substances in humans. We demonstrated that some perfluoroalkyl sulfonates are transported by NTCP; however, little was known about carboxylates. The purpose of this study was to determine if perfluoroalkyl carboxylates would interact with NTCP and potentially act as substrates. Sodium-dependent transport of [$^3$H]-taurocholate was measured in human embryonic kidney cells (HEK293) stably expressing NTCP in the absence or presence of perfluoroalkyl carboxylates with varying chain lengths. PFCAs with 8 (PFOA), 9 (PFNA), and 10 (PFDA) carbons were the strongest inhibitors. Inhibition kinetics demonstrated competitive inhibition and indicated that PFNA was the strongest inhibitor followed by PFDA and PFOA. All three compounds are transported by NTCP, and kinetics experiments revealed that PFOA had the highest affinity for NTCP with a K$_m$ value of 1.8 ± 0.4 mM. The K$_m$ value PFNA was estimated to be 5.3 ± 3.5 mM and the value for PFDA could not be determined due to limited solubility. In conclusion, our results suggest that, in addition to sulfonates, perfluorinated carboxylates are substrates of NTCP and have the potential to interact with NTCP-mediated transport.

**Keywords:** perfluoroalkyl acids; perfluoroalkyl carboxylates; taurocholate transport

## 1. Introduction

Perfluoroalkyl and polyfluoroalkyl substances (PFASs) are fluorinated fatty acid-like molecules that have been used in a variety of industrial applications [1]. Among them, perfluoroalkyl carboxylates (PFCAs) are a subset of PFAS compounds that do not undergo further degradation, hence they are often considered as the end-stage metabolites of the related precursor chemistries [2]. PFCAs are persistent due to chemical stability [3–5] and some of them can be detected in humans at low parts per billion levels [6–8]. Although the exact exposure routes have not been identified in the general population, dietary ingestion (food or water) has been suggested as the primary source of exposure to certain PFAS compounds [9].

Depending on the perfluoroalkyl chain length, once absorbed, some of the longer-chain PFCAs, such as perfluorooctanoic acid (perfluorooctanoate, PFOA), are slowly excreted in urine and/or feces in most of the species evaluated. Consequently, the serum elimination half-lives for PFOA can range from days to weeks in laboratory mice and non-human primates, or, several years in humans [7]. In contrast, shorter chain PFCAs, such as perfluorobutanoic acid (perfluorobutyrate, PFBA), can be rapidly eliminated via urine and the corresponding serum elimination half-lives range from a few hours in laboratory rodents to a few days in humans [7,10]. For these observed toxicokinetic differences among

various PFCA molecules (chain length and species dependency), one proposed underlying mechanism is that some of the PFCAs, such as PFOA, can be trapped in the enterohepatic circulation, meaning that PFOA along with bile and bile salts, can be secreted into the small intestine and subsequently reabsorbed and transported back to the liver [11–14]. This proposed mechanism of PFOA accumulation in the liver is further supported by reports of PFOA being preferential partitioned into serum and liver in laboratory animals [7,15].

The enterohepatic circulation of bile acids has been well characterized and it is known that several transport proteins in hepatocytes and enterocytes are crucial for this process. In human hepatocytes, the $Na^+$/taurocholate cotransporting polypeptide (NTCP) mainly mediates the sodium-dependent uptake of conjugated bile acids into hepatocytes [16], while the unconjugated bile acids are mainly transported by several organic anion transporting polypeptides (OATPs) [17]. We previously demonstrated that OATP1B1, OATP1B3, and OAP2B1 can transport selected PFAS compounds, including perfluorobutane sulfonate (PFBS), perfluorohexane sulfonate (PFHxS), perfluorooctane sulfonate (PFOS), as well as perfluorooctanoic acid (PFOA, C8) and perfluorononanoic acid (PFNA, C9) [18]. In addition, we have also reported that human NTCP can transport the perfluoroalkyl sulfonates PFBS, PFHxS, and PFOS [19]. However, it is largely unknown whether any of the PFCAs are actively transported by NTCP and to what extent PFCAs with different chain lengths would interact with human NTCP. In the present study, we investigated chain length-dependent inhibition of NTCP-mediated taurocholate uptake, performed inhibition kinetics, and measured direct uptake of inhibiting PFCAs by NTCP.

## 2. Materials and Methods

### 2.1. Materials

Radiolabeled [$^3$H]-taurocholate was purchased from PerkinElmer (Boston, MA, USA). Perfluoroalkyl carboxylates (C3–C18) were obtained from Sigma-Aldrich (St. Louis, MO, USA).

### 2.2. Cell Culture and Uptake Experiments

Flp-In™-293 (HEK293) cells were purchased from Thermo Fisher Scientific (Waltham, MA, USA) and grown at 37 °C in a humidified 5% $CO_2$ atmosphere in Dulbecco's Modified Eagle's Medium (DMEM) obtained from the American Type Culture Collection (ATCC, Manassas, VA, USA: 30-2002). The medium was supplemented with 10% fetal bovine serum (FBS) (Hyclone, Logan, UT, USA), and 100 U/mL penicillin, 100 μg/mL streptomycin (Thermo fisher Scientific). The pcDNA5/FRT plasmid containing the human NTCP open reading frame with a C-terminal 6-His tag [19] was used to generate a stable NTCP-expressing cell line using hygromycin selection. A single clone was isolated using a cloning cylinder and cultivated in the above DMEM with the addition of 500 μg/mL hygromycin (Thermo Fisher Scientific). For transport experiments, cells were plated on poly-D-lysine-coated 24-well plates at 300,000 cells per well for uptake experiments after 48 h or at 200,000 cells per well for uptake experiments after 72 h. Uptake experiments and LC-MS/MS analysis were performed at the time points indicated in the figure legends as previously described [19,20].

### 2.3. Statistical Analysis

Data were analyzed for significant differences using one-way ANOVA followed by "Dunnett's multiple comparisons test" using Prism 9 (GraphPad Software Inc., San Diego, CA, USA). A $p$-value smaller than 0.05 was considered significant.

## 3. Results

### 3.1. PFCAs Can Inhibit Taurocholate Uptake Mediated by the Human $Na^+$/Taurocholate Cotransporting Polypeptide (NTCP)

To investigate whether PFCAs would interact with the human $Na^+$/taurocholate co-transporting polypeptide (NTCP), we determined to what extent perfluorinated carboxylic

acids with chain lengths from 3 to 18 carbon atoms (C3 to C18) would inhibit NTCP-mediated taurocholate uptake. For these experiments, we measured the uptake of 1 μM taurocholate in the presence and absence of 10 or 100 μM of each of the PFCAs. As demonstrated in Figure 1, there is a chain length-dependent inhibition of sodium-dependent taurocholate (1 μM) uptake in HEK293 cells stably expressing NTCP. Perfluorodecanoic acid (C10; PFDA) and perfluorononanoic acid (C9; PFNA) had the strongest effect and inhibited NTCP-mediated uptake by 60–70% at the 10 μM concentration (Figure 1A). Perfluorooctanoic acid (C8; PFOA) and perfluoroundecanoic acid (C11) were also statistically significant inhibitors at 10 μM, but to a lesser extent. At a concentration of 100 μM, all the tested PFCAs, with the exception of perfluorobutanoic acid (C4), inhibited NTCP-mediated uptake of taurocholate (Figure 1B).

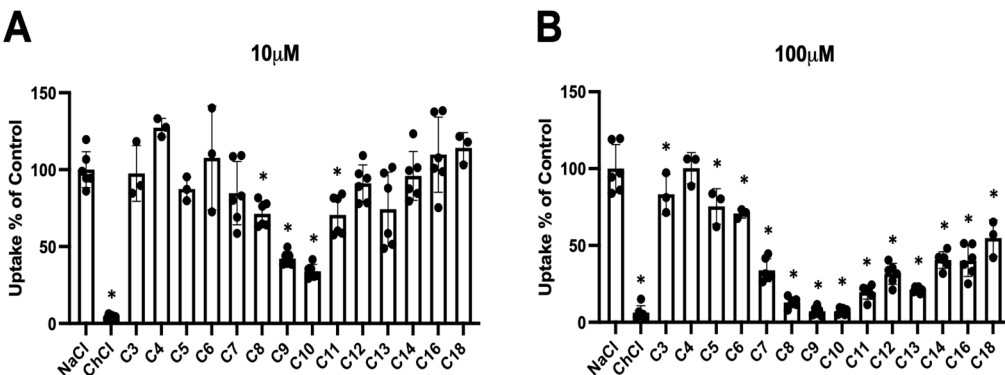

**Figure 1.** Chain length-dependent inhibition of NTCP-mediated taurocholate uptake by different PFCAs. Uptake of 1 μM [$^3$H]-taurocholate into NTCP-expressing HEK293 cells was inhibited by 10 (**A**) or 100 μM (**B**) PFCAs with increasing chain lengths of 3 (C3) to 18 (C18) carbon atoms. Uptake was measured in sodium-containing (NaCl) or sodium-free (ChCl) buffer for 1 min at 37 °C and corrected for protein. Uptake then was normalized to control (NaCl) and individual values of 1 or 2 experiments performed in triplicates are presented. Bars represent the mean ± SD, asterisks indicate significant differences to the control (NaCl) at $p < 0.05$.

### 3.2. Determination of Half-Maximal Inhibitory Concentrations for PFOA, PFNA, and PFDA

We next wanted to quantify the inhibitory potency of the three strongest inhibitors: PFOA, PFNA, and PFDA, by determining their IC$_{50}$ values. For these experiments, NTCP-mediated uptake of 1 μM taurocholate was measured in the absence and presence of increasing concentrations of perfluorinated compounds. For PFOA and PFNA, the concentrations ranged from 0.1 to 1000 μM, while the concentrations for PFDA were between 0.1 and 100 μM, due to solubility limitations. The results shown in Figure 2 demonstrate that all three compounds inhibited NTCP-mediated uptake, with IC$_{50}$ values in the lower micromolar range. The strongest inhibitors were PFNA and PFDA, with IC$_{50}$ values of 7.5 and 9.5 μM, respectively. Mirroring the results in Figure 1A, PFOA was the weakest inhibitor, with an IC$_{50}$ of 28 μM.

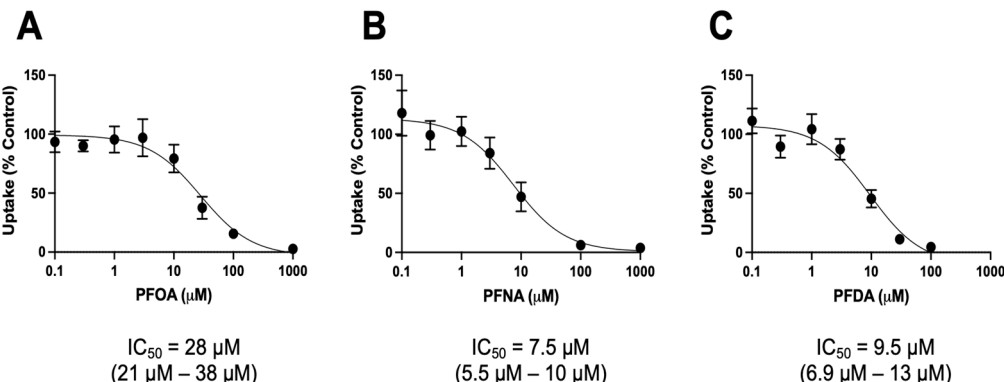

**Figure 2.** Concentration-dependent inhibition of NTCP-mediated taurocholate uptake by PFOA, PFNA, and PFDA. Uptake of 1 μM [³H]-taurocholate was measured into NTCP-expressing HEK293 cells in the absence or presence of increasing concentrations of (**A**) perfluorooctanoic acid (PFOA), (**B**) perfluorononanoic acid (PFNA), or (**C**) perfluorodecanoic acid (PFDA). Uptake was measured for 1 min at 37 °C in a sodium-containing buffer and corrected for protein. After conversion of the data to percent of control, IC₅₀ values were calculated using GraphPad Prism V9 using the "One site-Fit logIC₅₀" equation from 3 independent experiments performed in triplicates. The 95% confidence intervals are given in parentheses.

### 3.3. Inhibition Kinetics of NTCP-Mediated Taurocholate Transport for PFOA, PFNA, and PFDA

To determine the type of inhibition demonstrated in Figures 1 and 2, we performed inhibition kinetics by measuring the transport of increasing concentrations of tauro-cholate inhibited by 0, 10, and 100 μM of either PFOA, PFNA, or PFDA. As shown in Figure 3 and summarized in Table 1, inhibition of NTCP-mediated taurocholate uptake by the three PFCAs was competitive. The calculated $K_i$ values showed the same order as the IC₅₀ values, with PFDA having the lowest $K_i$ (8.3 ± 0.8 μM), followed by PFNA (12.3 ± 1.4 μM) and PFOA (17 ± 1.9 μM). This competitive inhibition suggested that the three PFCAs could be transported by NTCP.

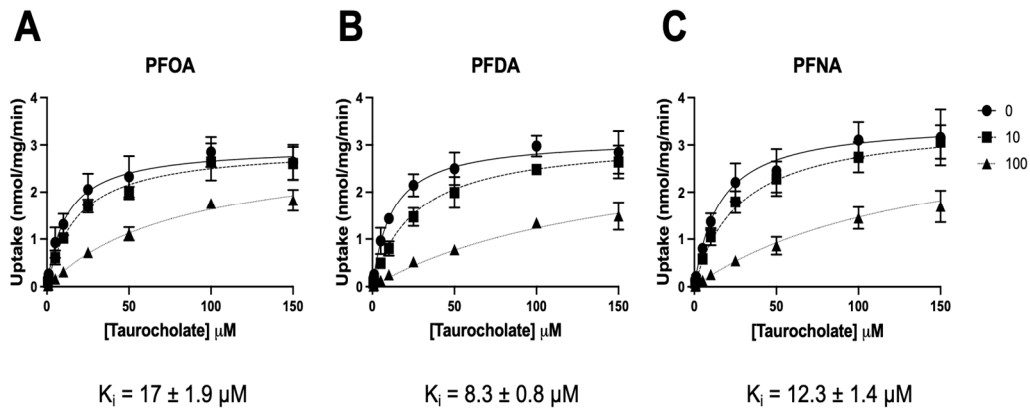

**Figure 3.** Kinetics of taurocholate uptake in the absence and presence of PFOA, PFNA, and PFDA. Uptake of increasing concentrations of [³H]-taurocholate into NTCP-expressing HEK293 cells was measured for 15 s (within the initial linear rate) at 37 °C in a sodium-containing buffer in the absence or presence of 10 and 100 μM of (**A**) PFOA, (**B**) PFDA, or (**C**) PFNA. After correction for protein, the results were analyzed using GraphPad Prism V9 using the "Enzyme kinetics–Inhibition" equations. Analysis for competitive inhibition resulted in the best fit. Mean ± SD values of two independent experiments performed in triplicate are shown.

**Table 1.** Kinetic parameters of NTCP-mediated taurocholate uptake and inhibition by PFOA, PFNA, and PFDA.

|  | Concentration of PFCA | $K_m$ (µM) | $V_{max}$ (nmol/mg·min) | $K_i$ (µM) |
|---|---|---|---|---|
| **PFOA** | 0 | 12 (8.7–16) | 2.9 (2.7–3.2) | |
| | 10 µM | 20 (16–25) | 3.0 (2.8–3.2) | 17 (14–21) |
| | 100 µM | 73 (60–92) | 2.8 (2.6–3.1) | |
| **PFDA** | 0 | 12 (9.4–15) | 3.2 (2.9–3.4) | |
| | 10 µM | 29 (23–36) | 3.2 (3.0–3.4) | 8.3 (6.9–10) |
| | 100 µM | 99 (73–139) | 2.5 (2.2–3.0) | |
| **PFNA** | 0 | 16 (12–22) | 3.5 (3.2–3.8) | |
| | 10 µM | 24 (19–30) | 3.5 (3.2–3.7) | 12 (10–15) |
| | 100 µM | 118 (78–192) | 3.1 (2.5–4.1) | |

Kinetic parameters were calculated using the Michaelis–Menten equation in GraphPad Prism V9 based on competitive inhibition by the PFCAs. Values in parenthesis show the 95% confidence intervals.

### 3.4. Time Dependency of NTCP-Mediated PFOA, PFNA, and PFDA Uptake

To resolve whether PFOA, PFNA, and PFDA were substrates of NTCP, we measured the transport of these compounds. Preliminary results performed at 10 µM demonstrated that all three PFCAs were actively transported by NTCP. To better characterize this transport, we wanted to perform kinetics experiments and determine the $K_m$ and $V_{max}$ values for the three compounds. Therefore, in Figure 4, we determined the initial linear rate conditions in which to perform the kinetics experiments. We evaluated the time-dependent uptake of PFOA, PFNA, and PFDA at 10 and 300 µM in NTCP-expressing HEK293 cells in sodium-containing and sodium-free buffer. While there is a time-dependent uptake of all three PFCAs, most likely due to organic anion transporters expressed in HEK293 cells, NTCP only transports its substrates in the presence of sodium. Therefore, we calculated the net sodium-dependent uptake. Based on the time-dependent sodium-dependent uptake results shown in Figure 4, we used 20 s for PFOA (Figure 4A), 30 s for PFNA (Figure 4B), and 1 min for PFDA (Figure 4C) to perform the concentration-dependent uptake kinetics.

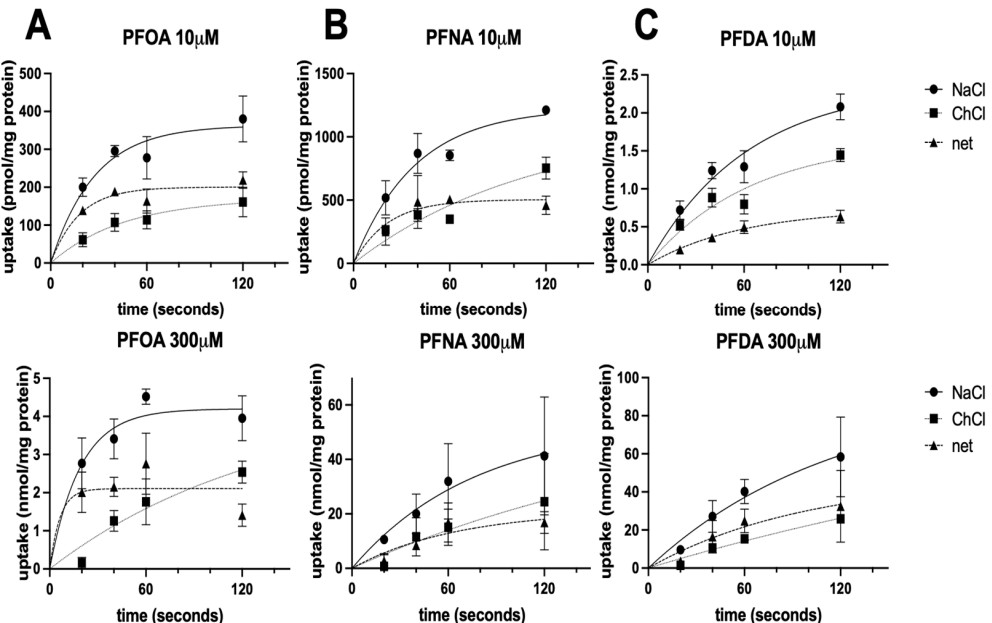

**Figure 4.** Time dependency of NTCP-mediated PFOA, PFNA, and PFDA uptake. Uptake of (**A**) perfluorooctanoic acid (PFOA), (**B**) perfluorononanoic acid (PFNA), or (**C**) perfluorodecanoic acid (PFDA) was measured in NTCP-expressing HEK293 cells in sodium-containing (NaCl; solid lines) or sodium-free (ChCl; dotted lines) buffer at the indicated times at 37 °C and corrected for protein. The net sodium-dependent uptake (net; dashed lines) is also shown. The results are presented as means ± SD of two combined independent experiments performed in triplicates.

### 3.5. Kinetics of NTCP-Mediated PFOA and PFNA Uptake

Uptake of sodium-dependent NTCP-mediated PFOA was saturable, with a $K_m$ value of $1.8 \pm 0.4$ mM and a $V_{max}$ value of $128 \pm 14$ nmol/mg/min (Figure 5A), clearly demonstrating that PFOA is a substrate of human NTCP. For the concentration-dependent uptake of PFNA, we were limited by solubility and could only determine uptake up to a concentration of 1 mM. As shown in Figure 5B, sodium-dependent PFNA uptake did not reach saturation but could be fitted to the Michaelis–Menten equation, thus a $K_m$ value of $5.3 \pm 3.5$ mM as well as a $V_{max}$ value of $670 \pm 391$ nmol/mg/min could be calculated. Uptake of sodium-dependent PFDA did not reach saturation within the concentration range that we could test (up to 300 mM). Therefore, no kinetic constants could be calculated for PFDA.

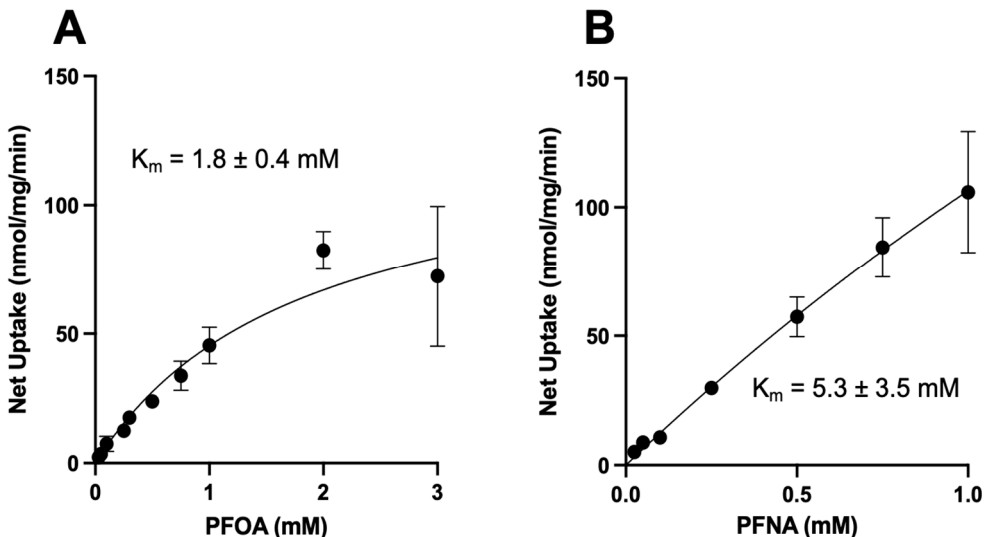

**Figure 5.** Kinetics of NTCP-mediated PFOA and PFNA uptake. Uptake of increasing concentrations of (**A**) perfluorooctanoic acid (PFOA) or (**B**) perfluorononanoic acid (PFNA) was measured at 37 °C for 20 (PFOA) or 30 s (PFNA) in NTCP-expressing HEK293 cells in sodium-containing or sodium-free buffer and corrected for protein. Net uptake was calculated by subtracting uptake in the sodium-free medium from uptake in the sodium-containing medium. Values are mean $\pm$ SD of three independent experiments performed in triplicates. Curves were fit using the Michaelis–Menten equation in GraphPad Prism V9.

## 4. Discussion

We have previously determined that certain perfluoroalkyl acids are transported by hepatic transporters. This includes the uptake of three perfluoroalkyl sulfonates (PFBS, PFHxS, and PFOS) by human NTCP [19] as well as by OATP1B1, OATP1B3, and OAP2B1 [18]. However, the transport of perfluorinated carboxylates by human NTCP had yet to be evaluated. We undertook this study to address (1) whether or not perfluoroalkyl carboxylates interacted with human NTCP or whether NTCP's interaction and transport of perfluoroalkyl acids would be limited to perfluoroakyl sulfonates, and (2) if NTCP and PFCAs did interact, to what extent would this interaction occurred and whether any of these compounds also served as substrates of NTCP.

Within the limit of the current study design where NTCP was stably expressed in HEK293 cells, we observed that perfluoroalkyl carboxylates indeed inhibit taurocholate uptake mediated by NTCP, and some are transported substrates. Initial experiments demonstrated a chain length-dependent inhibition of NTCP-mediated uptake of taurocholate (Figure 1). The strongest inhibitors were PFOA, PFNA, and PFDA and the degree of inhibition increased with chain lengths. This chain length-dependent inhibition was previously seen for the perfluoroalkyl sulfonates, with PFOS (an eight-carbon perfluoroalkyl

sulfonate) being the strongest inhibitor and PFBS (a four-carbon perfluoroalkyl sulfonate) being the weakest inhibitor of the compounds tested [19].

A simple explanation for this repeated chain length-dependent inhibition might be similarities between the molecular mass of these compounds and the model NTCP substrate, taurocholate. Stronger inhibitors like PFNA, PFDA, and PFOS have anionic molecular weights of 463.06, 513.08, and 499.13 g/mol, respectively, all of which are close to the molecular weight of taurocholate, 515.7 g/mol, potentially leading to competitive binding within the taurocholate docking site within NTCP. Another explanation is that although structurally different, the hydrophobic PFCAs might align in the substrate binding pocket or translocation pathway of NTCP, where the hydrophobic moiety of the amphipathic taurocholate would interact. However, the potential interaction between these PFCAs and NTCP can be further explored via 3-D docking modelling once the NTCP crystal structure becomes available in the future.

All three compounds (PFOA, PFNA, and PFDA) are competitive inhibitors of NTCP-mediated taurocholate uptake (Figure 3). As expected, the order of their $K_i$ values was the same as their $IC_{50}$ values, given that all three compounds are competitive inhibitors. These results also suggested that the three perfluoroalkyl carboxylates would likely be transported by NTCP. As mentioned, we confirmed that all three are substrates of NTCP and we could determine apparent affinity constants for PFOA and PFNA. However, the latter results are a rough estimate given kinetics saturation could not be reached due to solubility limitations. In addition, we were unable to determine a $K_m$ value for PFDA again due to limited substrate solubility. Interestingly, the $K_i$ value of PFOA is about 100-fold lower than its $K_m$ value. We are not aware of other competitive NTCP inhibitors with similar differences in their $K_i$ and $K_m$ values, but a potential explanation would be that while PFOA binds with high affinity to the transporter, its translocation is not very efficient, resulting in a much higher $K_m$ value.

Comparison of the perfluoroalkyl carboxylates $K_m$ values to the $K_m$ values previously obtained with perfluoroalkyl sulfonates (39.6 μM for PFBS, 112 μM for PFHxS, and 130 μM for PFOS [19]) clearly demonstrates that NTCP has a much higher affinity for the sulfonates than the carboxylates. We speculate that this increased affinity could be due to the presence of similar sulfur-containing groups in PFASs, in part, when considering the presence of the sulfate group in several known NTCP substrates: estrone-3-sulfate, dehydroepiandrosterone sulfate (DHEAS), pregnenolone sulfate, T3 sulfate, T4 sulfate as well as bromosulfophthalein [21–23]. Therefore, it is likely that additional sulfonated compounds are also transported by NTCP.

What do these results mean with respect to the disposition of perfluoroalkyl carboxylates and their impact on the function of NTCP in humans? The estimated mean serum PFOA level in occupational workers was 4.3 μM (1.78 ppm) [24]. In community residents whose drinking water contained higher levels of PFOA (and prior to granulated activated carbon filter installation), the estimated mean serum PFOA concentration was 0.9 μM (0.374 ppm) [25]. Compared to the $IC_{50}$ and $K_i$ values, it seems that PFOA levels in these subjects should not affect NTCP-mediated bile acid uptake into hepatocytes. Therefore, it seems unlikely that NTCP-mediated uptake of PFCAs plays a major role in the mechanisms that lead to the long elimination half-lives of these carboxylates in humans [26–28].

In summary, this study has shown that human NTCP can transport PFOA, PFNA, and PFDA and that all three perfluoroalkyl carboxylates are competitive inhibitors of NTCP-mediated uptake of taurocholate, with $K_i$ values in the lower micromolar range. These results demonstrate that NTCP has a wider substrate specificity than previously observed: in addition to several perfluoroalkyl sulfonates, this study has demonstrated that NTCP also transports perfluoroalkyl carboxylates. Based on the low affinities, we conclude that it is unlikely that these compounds will affect NTCP-mediated bile acid or drug uptake, even at the concentrations documented in occupational workers.

**Author Contributions:** Conceptualization, M.J.R. and B.H.; methodology, M.J.R., H.M., J.Y.I. and J.D.Z.; formal analysis, M.J.R., H.M., J.Y.I. and B.H.; investigation, M.J.R., H.M., J.Y.I. and J.D.Z. (provided LC-MS/MS analytical support); resources, S.-C.C. and B.H.; data curation, M.J.R. and B.H.; writing—original draft preparation, B.H.; writing—review and editing, M.J.R., S.-C.C. (background information on PFAS literature only) and B.H.; visualization, M.J.R. and B.H.; supervision, B.H.; project administration, B.H.; funding acquisition, B.H. All authors have read and agreed to the published version of the manuscript.

**Funding:** This research was funded by National Institute of Health grants P20 GM103549 and R01 GM077336, and by an unrestricted research grant from the 3M Company.

**Institutional Review Board Statement:** Not applicable.

**Informed Consent Statement:** Not applicable.

**Data Availability Statement:** All data are contained within the article.

**Conflicts of Interest:** J.D.Z. and S.-C.C. are employees of the 3M company.

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
