# Peer review of "Perfluoroalkyl Carboxylic Acids Interact with the Human Bile Acid Transporter NTCP"

_livers, doi:10.3390/livers1040017_

Round 1

Reviewer 1 Report

In this manuscript, Ruggiero et al reported that in addition to OATPs, PFOA, PFNA and PFDA can be transported by human NTCP.  Overall, the studies are well designed and conducted. The manuscript is well presented.  Some comments:

  1. Figure 3: PFOA, PFDA and PFNA at 10microM showed apparent competitive inhibition of NTCP uptake. However, PFOA, PFDA and PFNA at 10microM at 100microM did not show clear completive inhibition pattern.  Is this simply due to too high concentration of PFCAs used? This should be addressed.
  2. Figure 4: in sodium-free buffer, PFCAs can still be uptaked to some extent. This should be addressed.
  3. In the Discussion, the authors suspect that size similarity between PFNA/PFDA/PFOS and taurocholate is one potential explanation why PFCAs are competitive inhibitors of NTCP. However, structurally, PFCAs and taurocholate are largely different, which has not been discussed.  This should be addressed.
  4. To be consistent with other figures, Fig. 3B (10-carbon) and Fig, 3C (9-carbon) should be switched.

Author Response

We would like to thank the reviewers for their thoughtful comments and suggestions that helped to improve the manuscript.

Reviewer 1:

Comments and Suggestions for Authors

In this manuscript, Ruggiero et al reported that in addition to OATPs, PFOA, PFNA and PFDA can be transported by human NTCP.  Overall, the studies are well designed and conducted. The manuscript is well presented.  Some comments:

  1. Figure 3: PFOA, PFDA and PFNA at 10microM showed apparent competitive inhibition of NTCP uptake. However, PFOA, PFDA and PFNA at 10microM at 100microM did not show clear completive inhibition pattern.  Is this simply due to too high concentration of PFCAs used? This should be addressed. We are sorry but we did not understand the reviewers comment. When we used the data shown in Figure 3 to fit the Michaelis-Menten equation the Km values increased with increasing concentrations of PFCAs used while the Vmax values did not change. The calculated numbers are presented in Table 1. Such an increase in Km without a change in Vmax means that the inhibition is competitive.
  2. Figure 4: in sodium-free buffer, PFCAs can still be uptaked to some extent. This should be addressed. We have added the following text to the description of Figure 4:” While there is a time-dependent uptake of all three PFCAs, most likely due to organic anion transporters expressed in HEK293 cells, NTCP only transports its substrates in the presence of sodium. Therefore, we calculated net, sodium-dependent uptake. Based on the time-dependent sodium-dependent uptake results shown in Figure 4, we used 20 seconds for PFOA (Figure 4A), 30 seconds for PFNA (Figure 4B) and 1 minute for PFDA (Figure 4C) to perform the concentration dependent uptake kinetics.”
  3. In the Discussion, the authors suspect that size similarity between PFNA/PFDA/PFOS and taurocholate is one potential explanation why PFCAs are competitive inhibitors of NTCP. However, structurally, PFCAs and taurocholate are largely different, which has not been discussed.  This should be addressed. We added the following statement to the discussion: “Another explanation is that although structurally different, the hydrophobic PFCAs might align in the substrate binding pocket or translocation pathway of NTCP where the hydrophobic moiety of the amphipathic taurocholate would interact.” 
  4. To be consistent with other figures, Fig. 3B (10-carbon) and Fig, 3C (9-carbon) should be switched. We thank the reviewer for this suggestion and have switched the order in Figure 3.

Reviewer 2 Report

This is a very clear and well performed study of the interaction of perfluoroalkyl carboxylic acids with the human liver bile acid transporter, NTCP. I had only a few minor comments/questions.

1. Figure 4. The graph titles say PFOA 10 µM combined, PFNA 10 µM combined, etc. But panel C does not say combined. I think I know what the authors mean by "combined" but please clarify.

2. Figure 5. The units - µM versus mM do not agree in the figure and results section. Please correct. I think the authors mean mM for PFOA in the text.

3. Please comment on why the Ki values are so much lower (more potent) versus the Km values for transport. The values differ about 100-fold. How does this compare for other competitive inhibitors or NTCP? 

Author Response

This is a very clear and well performed study of the interaction of perfluoroalkyl carboxylic acids with the human liver bile acid transporter, NTCP. I had only a few minor comments/questions.

  1. Figure 4. The graph titles say PFOA 10 µM combined, PFNA 10 µM combined, etc. But panel C does not say combined. I think I know what the authors mean by "combined" but please clarify. Thanks for catching this. We did combine the two experiments but did not include this in all the titles. We removed “combined” from the figure and added this information in the legends to the figure.
  2. Figure 5. The units - µM versus mM do not agree in the figure and results section. Please correct. I think the authors mean mM for PFOA in the text. This was corrected in the text to mM.
  3. Please comment on why the Ki values are so much lower (more potent) versus the Km values for transport. The values differ about 100-fold. How does this compare for other competitive inhibitors or NTCP? The fact that the Ki values are so much lower than the Km values suggest that PFCAs have a high affinity for NTCP and can efficiently prevent taurocholate from being transported but are themselves not very good transported substrates. In order to calculate Km values the substrate has to be transported across the cell membrane and its intracellular concentration (after lysis of the cells) will be measured. Therefore, if the translocation step is very slow (or not very efficient) this would result in a high Km value. We are not aware of many other competitive inhibitors where Ki values for inhibition of taurocholate uptake and the corresponding Km values would be available. One exception is a paper by Greupink et al. 2011 where fluvastatin was characterized as a competitive inhibitor of NTCP-mediated taurocholate uptake and also as a substrate of NTCP. The reported IC50 value was 40uM and the corresponding Km value was 250uM, thus qualitatively also a higher Km value compared to the IC50 but not by the same magnitude as what we have seen with the PFCAs. We have added the following statement to the discussion:” Interestingly, the Ki value of PFOA is about 100-fold lower than its Km value. We are not aware of other competitive NTCP inhibitors with similar differences in their Ki and Km values but a potential explanation would be that while PFOA binds with high affinity to the transporter, its translocation is not very efficient resulting in a much higher Km value.”

Reviewer 3 Report

In this manuscript, Ruggiero and others present their results that PFOA, PFNA and PFDA inhibit taurocholate transport by NTCP. They have previously shown that OATP can transport selected PFAS compounds, including PFOA etc, and they want to know whether NTCP can also transport PFCA or not. To test their hypothesis, they have performed some biochemical assays.

1. As the authors stated in the Introduction, in human hepatocytes, NTCP mainly mediates the sodium-dependent uptake of conjugated bile acids into hepatocytes, while the unconjugated bile acids are mainly transported by several OATPs. There are several professors doing liver research in the authors' institute, is it possible to repeat at least one of the core experiments using human hepatocytes, which should be more physiological relavant?

2. Related to the above comment, one of the most important things is, all the experiments here are in vitro, using the HEK293 cells expressing NTCP on the plasmid. This is so far away from what happens in the real human body. The Reviewer understand that it might not be feasible to do animal work, but the authors should make it clear when they draw the conclusion. Everything is only based on the in vitro biochemical assays, so the statements should not, and cannot, be too strong or too conclusive. For instance, the Km here is really high, which means the interaction between PFCAs and NTCP is very weak, if there is any. But that is not necessarily true in vivo, because there are so many other factors in vivo which might help this interaction. Apparently those potentially factors are missing here and this possibility cannot be ruled out.

3. Results part 3.1, title said "PFCAs interact with NTCP". The authors did not provide any direct evidence showing that PFCAs really, physically interact with NTCP. This title is misleading and not accurate. Based on Figure 1, one can only say that PFCAs affect NTCP-mediated taurocholate transportation. The authors should change the title.

4. There is a mistake in the legend of Figure 1. It should be p<0.05, not 0.5

5. Are there any other molecules contribute to the taurocholate transportation here in the assays? If not, why the ChCl controls are so different for PFOA, PFNA and PFDA in Figure 4? Theoritically they should be similar, correct? In other words, did the authors have such controls, which only used HEK293 cells without NTCP, to see if PFCAs still inhibit taurocholate uptake? Seems that the HEK293 cells are able to uptake taurocholate (ChCl controls, as the NTCP uptake is sodium-dependent), based on the results in Figure 4. This control should also be applied to Figure 1.

6. In the text in part 3.5, the units for the Km values should be mM, not uM. It's not consistent with Figure 5, and the Reviewer believe that it's a mistake in the main text. Please correct it.

Author Response

In this manuscript, Ruggiero and others present their results that PFOA, PFNA and PFDA inhibit taurocholate transport by NTCP. They have previously shown that OATP can transport selected PFAS compounds, including PFOA etc, and they want to know whether NTCP can also transport PFCA or not. To test their hypothesis, they have performed some biochemical assays.

  1. As the authors stated in the Introduction, in human hepatocytes, NTCP mainly mediates the sodium-dependent uptake of conjugated bile acids into hepatocytes, while the unconjugated bile acids are mainly transported by several OATPs. There are several professors doing liver research in the authors' institute, is it possible to repeat at least one of the core experiments using human hepatocytes, which should be more physiological relavant? Please see response to point 2.
  2. Related to the above comment, one of the most important things is, all the experiments here are in vitro, using the HEK293 cells expressing NTCP on the plasmid. This is so far away from what happens in the real human body. The Reviewer understand that it might not be feasible to do animal work, but the authors should make it clear when they draw the conclusion. Everything is only based on the in vitro biochemical assays, so the statements should not, and cannot, be too strong or too conclusive. For instance, the Km here is really high, which means the interaction between PFCAs and NTCP is very weak, if there is any. But that is not necessarily true in vivo, because there are so many other factors in vivo which might help this interaction. Apparently those potentially factors are missing here and this possibility cannot be ruled out. We agree that there are limitations to the presented study and to highlight that these are results obtained with HEK293 cells and not hepatocytes, we have modified the discussion in lines 211/212 as follows: “Within the limit of the current study design where NTCP was stably expressed in HEK293 cells, we observed that perfluoroalkyl carboxylates indeed inhibit taurocholate uptake mediated by NTCP, and some are transported substrates.” Furthermore, we completely agree with the reviewer’s comment that “the Km here is really high, which means the interaction between PFCAs and NTCP is very weak, if there is any”. Therefore, we do not believe that experiments using human hepatocytes would add to these findings given that the conclusions are limited to NTCP and state “, it seems unlikely that NTCP-mediated uptake of PFCAs plays a major role in the mechanisms that lead to the long half-life of these carboxylates in humans“ (lines 258-260) and that “it is unlikely that these compounds will affect NTCP-mediated bile acid or drug uptake” (lines 267/268).
  3. Results part 3.1, title said "PFCAs interact with NTCP". The authors did not provide any direct evidence showing that PFCAs really, physically interact with NTCP. This title is misleading and not accurate. Based on Figure 1, one can only say that PFCAs affect NTCP-mediated taurocholate transportation. The authors should change the title. We changed the title of part 3.1. to “PFCAs can inhibit taurocholate uptake mediated by the human Na+/taurocholate cotransporting polypeptide (NTCP)”.
  4. There is a mistake in the legend of Figure 1. It should be p<0.05, not 0.5 We thank the reviewer for catching this mistake. It was corrected.
  5. Are there any other molecules contribute to the taurocholate transportation here in the assays? If not, why the ChCl controls are so different for PFOA, PFNA and PFDA in Figure 4? Theoritically they should be similar, correct? In other words, did the authors have such controls, which only used HEK293 cells without NTCP, to see if PFCAs still inhibit taurocholate uptake? Seems that the HEK293 cells are able to uptake taurocholate (ChCl controls, as the NTCP uptake is sodium-dependent), based on the results in Figure 4. This control should also be applied to Figure 1. We respectfully disagree with the statement that the ChCl controls for the three PFCAs should be the same. We would like to emphasize that in Figure 4 we did not measure taurocholate uptake but the uptake of the three different PFCAs. If we would have measured taurocholate uptake in the absence and presence of these compounds, we agree, we should have seen the same values for all three compounds in the ChCl control. The three different compounds have different chain lengths. Therefore, their hydrophobicity increases with increasing chain length (their calculated logP values increase with chain length) and therefore we would have expected that their unspecific binding to the cells, measured here in the ChCl condition, would increase, and it in fact does increase.
  6. In the text in part 3.5, the units for the Km values should be mM, not uM. It's not consistent with Figure 5, and the Reviewer believe that it's a mistake in the main text. Please correct it. We apologize for this mistake and have corrected the µM to mM in the text.

Round 2

Reviewer 3 Report

The authors have answered the questions from this Reviewer. Thanks.